# Risk of all-cause mortality associated with chronic obstructive pulmonary disease and the role of healthy ageing trajectories: a population-based study of middle-aged and older adults

Ivet Bayes-Marin ,[1,2,3] Albert Sanchez-Niubo ,[1,3] Daniel Fernández ,[4] Josep Maria Haro ,[1,2,3] Beatriz Olaya [1,3]

► Prepublication history and additional online supplemental material for this paper are available online. To view these files, please visit the journal online. To view these files, please visit the journal online (http://dx.doi.org/10.1136/bmjopen-2021-050947).

For numbered affiliations see end of article.

**Correspondence to**
Dr Albert Sanchez-Niubo; albert.sanchez@pssjd.org

## ABSTRACT

**Objectives** The aims were to study the risk of all-cause mortality associated with chronic obstructive pulmonary disease (COPD) and healthy ageing trajectories (HAT) in three birth cohorts and to determine the moderating role of HAT in the association between COPD and all-cause mortality.

**Design** Prospective cohort study.

**Setting** Data from waves 1 to 5 of The Survey of Health, Ageing and Retirement in Europe.

**Participants** The total sample was 28 857 community-dwelling individuals aged 50+ years.

**Main outcome** All-cause mortality associated with COPD and HAT adjusting for covariates. We performed Aalen additive hazards models to explore these associations. Interactions between COPD and HAT were also explored. Analyses were conducted separately in three birth cohorts (>1945, 1936–1945 and ≤1935). Latent class growth analysis was used to classify participants into HAT.

**Results** Three parallel HAT were found in the three birth cohorts ('low', 'medium' and 'high' healthy ageing). Participants with COPD had an increased mortality risk, but this effect was no longer significant after adjusting for covariates. The 'low' HAT was associated with increased mortality risk in the three subsamples, although this effect was lower after adjustment. The interaction between COPD and HAT was significant only in the ≤1935 birth cohort, indicating that those with COPD and a 'low' trajectory had a greater risk of mortality.

**Conclusions** The healthy ageing scale may be a suitable tool to identify patients at higher risk to mitigate disease burden and improve patients' quality of life.

## INTRODUCTION

Chronic obstructive pulmonary disease (COPD) is a major cause of morbidity and mortality worldwide.[1 2] COPD is expected to become the third leading cause of death by 2030.[3] The growing burden of COPD is a consequence of population ageing and the continued use of tobacco, which is considered its main risk factor.[4] Moreover, air pollution

### Strengths and limitations of this study

► The analyses were performed in different birth cohorts (>1945, 1936–1945 and ≤1935) to assess differences in mortality risks related to societal changes, such as lifestyle behaviours and occupation trends.

► We used a novel measurement scale of healthy ageing including intrinsic capacity and functional ability variables.

► The calculation of Aalen's additive hazards models rather than Cox models allowed the inclusion of time-variant variables in the analyses.

► Due to the high percentage of missingness in the age of the diagnosis of the diseases, we selected the age of the earliest diagnosis of each disease within the five waves.

► For the survival analysis, we used the age of the participants instead of the years of the interview for better interpretation. However, this introduces a problem of left truncation since the age range observed for each participant is different, although we took this into account in the additive regression model.

has been associated with acute exacerbations of COPD, increased respiratory morbidity and mortality.[5]

COPD is characterised by a progressive airflow limitation associated with an abnormal inflammatory response of the lungs to noxious particles or gases.[6] With a worsening of the disease, there can be an increase of functional ability limitations in the activities of daily living (ADL) and in the instrumental activities of daily living (IADL), limiting exercise performance and self-care.[7–9] COPD has increasingly been recognised as a multi-component disease, associated with a wide range of physical diseases and psychological disorders.[10] Non-communicable diseases

(NCDs) such as hypertension, cardiovascular diseases, diabetes, cancer and depression commonly coexist in patients with COPD, worsening its progression.[1 2 11] Furthermore, cognitive impairment is common among COPD patients, suggesting that impaired performance in neuropsychological tests might be a predictor of early mortality for people diagnosed with COPD.[10]

Despite being a growing public health concern, there is a lack of epidemiological data about the prevalence and distribution of COPD.[6 12] The paucity of information on COPD prevalence and incidence is partly due to differences in the methods used for its diagnosis and classification, often being underestimated.[6] These differences in the assessment methods and definitions have also hampered the comparison of COPD prevalence and impact across countries. Both The Burden of Lung Disease (BOLD) project[4] and the Latin American Project for the Investigation of Obstructive Lung Disease (PLATINO)[13] were developed to map the COPD prevalence using the same methodology in different countries. Those studies were performed in China and Turkey, and five Latin American countries, respectively. Nevertheless, there is a lack of integrated and updated estimates of COPD prevalence, similar to BOLD and PLATINO initiatives, and information regarding patient's quality of life impact and associated mortality in Europe.[2 12] Additionally, available studies suggest large differences across European countries regarding prevalence rates of COPD and the associated death rates.[14 15] In a systematic review, COPD prevalence ranged from 3% in Finnish women to 57% in Italian men and women.[14] Some differences have also been found in COPD-related mortality across European countries and between men and women.[15] Overall, regarding European countries, COPD-related mortality rates appeared to decline in men in most countries from 1995 to 2017, whereas mortality rates due to COPD increased in women from +2% per year in Austria to +4.2% or+4.8% per year in the Czech Republic and Hungary, respectively.[15]

In that sense, an integrated longitudinal dataset that considers different European countries could be particularly useful in studying the risk of mortality associated with COPD in different European countries. ln particular, using a cross-national panel database may prevent possible heterogeneities arising from differences in survey methodologies, diagnostic criteria and population structure.[16] The study of the mortality risk in COPD patients, and its association with several variables related to health and functioning would allow identifying those vulnerable sectors of the population and creating of preventive measures and interventions in diverse healthcare systems.

Previous studies focused on the association between exercise capacity and mortality among patients with COPD, which has been considered one of the best predictors of mortality.[17–19] Measures of exercise capacity include indicators such as body mass index, airflow obstruction, dyspnoea, handgrip strength and the sit-to-stand test.[20] Nevertheless, these indicators of exercise capacity are just measures of intrinsic capacity that do not capture the individual's functional ability over the life course. In that sense, the functional ability results from the interaction of the individuals' intrinsic capacity, including physical and mental capacities, and their environment, as access to medications, personal and assistive support or physical barriers.[21] Therefore, a measure assessing both intrinsic capacity and functional ability may be a better way to capture a person's healthy ageing.

Several authors advocate for using composite measures as the International Classification of Functioning, Disability and Health to assess COPD patients' complexity, including also functional capacity and functional performance.[22] Related to this, the Ageing Trajectories of Health: Longitudinal Opportunities and Synergies (ATHLOS) project[23] developed a healthy ageing scale[21] using 16 international cohort studies to determine the intrinsic capacity and functional ability of the participants allowing comparisons across countries. The healthy ageing scale comprises several domains such as vitality, sensory skills, locomotion/mobility, cognition, ADL and IADL. Thus, this measure includes measures of exercise capacity and functionality that could be affected by the course of COPD and impact on the patients' quality of life.

The aims of the present paper are: (1) to study the risk of all-cause mortality associated with COPD and healthy ageing trajectories (HAT) in three population-based cohorts of middle-aged and older adults and (2) to determine the moderating role of HAT in the association between COPD and all-cause mortality. We speculated that a HAT characterised by low levels of healthy ageing would be significantly associated with an increased risk of mortality in people with COPD. In contrast, individuals with higher levels of healthy ageing and COPD would have a lower risk of mortality.

## METHODS
### Study design and data collection
The present study used data from five waves of The Survey of Health, Ageing and Retirement in Europe (SHARE).[24] SHARE is a multidisciplinary, cross-national panel database that contains a broad range of information on health, socioeconomic status and social networks of European citizens aged 50 and older. The first wave took place in 2004–2005, constituted by more than 22 000 persons born in 1954 and earlier, and the following waves were conducted approximately every 2 years. The interviewers used computer-assisted personal interviewing (CAPI) to collect most of the data in all waves. Additionally, in waves 1, 2 and 4, self-administered questionnaires were handed out after the CAPI completion. If a respondent passed away during the study, then an end-of-life interview was conducted with a proxy.

The overall individual response rate at baseline was 60.1%, and the wave-to-wave retention rate of participants from wave 1 was higher than 55% in all the

countries.[25] All participants gave written consent. Ethical approvals for waves from 1 to 3 were granted by the Ethics Committee of the University of Mannheim.[24] For waves 4 and 5, the SHARE projects were reviewed and approved by the Ethics Council of the Max-Planck Society.[26] Further details concerning the study design of SHARE can be found elsewhere.[24]

The following countries were included in the present analysis: Denmark, Sweden, Greece, Italy, Spain, Israel, Austria, Belgium, France, Germany, Netherlands and Switzerland. We excluded participants incorporated in the subsequent waves due to the sample's refreshments (n=30 816). The analyses focused on people aged 50 years and older who completed a non-proxy interview at baseline, resulting in an analytical sample of 28 857 respondents.

### Patient and public involvement
No patient involved.

### Measurements
#### All-cause mortality
The death of a participant was confirmed by interviewing a proxy-respondent since information on the deceased was not linked to national death registries.[25 27] If confirmed, the date of death was obtained from end-of-life interviews with a proxy respondent.[25 27] Participants were characterised as survivors or censored if they were alive at the end of the study period, and dead if they died during the study period.

Survival time was calculated in years and as follows: (1) from baseline to the reported date of death or the final assessment date for those participants who were alive at the end of 2013 or (2) in the case that a participant reported being diagnosed with COPD at baseline, survival time was calculated from baseline. Besides, for the set of patients who reported a new diagnosis of COPD during the follow-up period, we considered the first time of the observation as the age at which they were newly diagnosed.

#### Chronic obstructive pulmonary disease
Participants reported whether a doctor ever informed them that they had 'COPD such as chronic bronchitis or emphysema'. In the present study, we considered the first age in which a participant reported having been diagnosed with COPD instead of considering the presence/absence of COPD at baseline because the participant might be diagnosed in the subsequent waves. Therefore, COPD diagnosis was treated as a time-variant variable.

#### Healthy ageing scale
We used an international scale of healthy ageing measurement developed by the ATHLOS consortium.[21 23] This scale used items about intrinsic capacity and functional ability based on the World Mental Health's (WHO) concept of healthy ageing.[28] The healthy ageing scale covers different domains, such as vitality, sensory skills, locomotion/mobility, cognition, ADL and IADL.

Thirty-nine study-specific variables were harmonised into dichotomous items indicating the presence or absence of difficulties (see online supplemental table 1). Final scores were estimated for all individuals and converted to T-scores with a mean of 50 and an SD of 10. We applied latent class growth analysis (LCGA)[29] to identify longitudinal trajectories according to the healthy ageing scale score across the waves and classify the participants into those trajectories.

### Covariates
Demographic variables included sex (male/female), age (in years), level of education (less than primary, primary, secondary, and tertiary), marital status (single, married or currently cohabiting, separated or divorced and widowed) and quintiles of household wealth (first quintile indicating lowest level).

Lifestyles and health behaviours included ever smoked and practice of vigorous physical activity during the last 2 weeks and were coded as yes or no. The following self-reported diagnoses of NCDs different from COPD were included: diabetes, hypertension, joint disorders (arthritis, rheumatism, or osteoarthritis), asthma, myocardial infarction and stroke. Similar to COPD, we selected the age of the earliest diagnosis of each NCD across the five waves, considering them as time-variant variables.

Depression was assessed with the EURO-D 12-item scale, which was developed and validated for the EURODEP studies to measure depressive symptoms across European countries, accounting for regional differences.[30 31] The EURO-D score ranges from 0 to 12, with higher scores meaning higher levels of depression, being four or greater than the proposed cut-off score that has been selected to create a dichotomous depression variable (yes/no).[30]

Finally, we grouped the countries into three European regions according to the WHO and the United Nations Statistical Division regional classification.[32 33] Thus, Northern Europe was constituted by Denmark and Sweden; Western Europe included Austria, Belgium, France, Germany, Israel, the Netherlands; and Switzerland, and Southern Europe included Spain, Italy and Greece.

### Statistical analyses
We divided the sample into three groups according to the participants' birth year and kept proportional sample sizes. The first group (n=9866) was composed of those participants who were born after 1945 (the youngest participants: aged 50+), the second group (n=9254) comprised participants born between 1936 and 1945 (ages from 58 to 70 years old) and the third one (n=9739) encompassed individuals who were born in 1935 or earlier (the oldest participants: from 69 to 104 years old). Analyses were independently conducted in these three birth cohorts.

LCGA was used to classify individuals into trajectories based on their score on the healthy ageing scale.[29] The number of trajectories was determined by analysing

group models from 1 to 5 trajectories. According to the Bayesian information criterion, the optimal model was selected. The lowest value indicates the better fit[34 35] and the sample size of the trajectory group. In addition, a sample size lower than 5% was considered insufficient to identify classes.[35]

To analyse the associations between COPD and time to death, we conducted an Aalen additive hazards modelling approach, avoiding the assumption of proportionality of the Cox regression hazards.[36 37] These models can provide a better picture of how the effects of covariates develop over time without assuming the proportional risk hypothesis as in the Cox regression models.[38] Parameters of these models are arbitrary cumulative regression functions that represent the cumulative excess risk at each unit of time and are useful to assess changes over time graphically.[39] CIs above zero for a concrete age indicate a significant risk, below zero indicate a protective effect, and CIs including zero show a non-significant risk.[40] Models were adjusted for sex, age, marital status, level of education, household wealth, region, vigorous physical activity, tobacco consumption, HAT, depression and presence of NCDs, all as time-varying covariates. The interaction between COPD and HAT was also assessed. Age was used as the time measure. Participants who were alive at the end of the study period or in their final assessment were censored. In the modelling process, data were left-truncated because we considered the first interview as the time of diagnosis in the participants whom and NCD had been diagnosed before baseline. All analyses were performed using R V.4.0.3.[41] Statistical significance was set at p<0.05.

## RESULTS

We identified three HAT in each of the three birth cohorts according to lower BIC and the sample sizes not lower than 5% (online supplemental table 2 and online supplemental figure 1). Although four trajectories met the selection criteria in the oldest birth cohort, we decided to select a three-trajectory model to facilitate comparison between the three cohorts. In all birth cohorts, the trajectories were parallel. The first trajectory group included individuals with the highest scores on the healthy ageing scale and the third with the worse scores. We named each trajectory group as 'high', 'medium' and 'low', respectively.

Table 1 shows the characteristics of participants. Those participants of the oldest group (born ≤1935) showed a higher prevalence of COPD (12.50%), followed by those born between 1935 and 1945 (9.57%), (p<0.001). The oldest group presented lower proportions of the 'high' HAT (31.60%) compared with the other two birth cohorts (p<0.001). Finally, the proportion of deaths increased with age, being lower in the >1945 (2.07%) and higher in the ≤1935 subsample (16.90%) (p<0.001).

Three Aalen regression models were conducted: one with only the variable COPD, the second with only the HAT variable, and the third with COPD and the HAT adjusted for covariates. The estimated cumulative coefficients of the first and second model are presented in online supplemental figure 2, and those from the third model are presented in figure 1, according to the >1945, 1936–1945 and ≤1935 subsamples, respectively. In the first model, COPD diagnosis had a significant risk on mortality in the three birth cohort groups: from 74 years old onwards in the ≤1935 subsample, from 65 years old onwards in the other two subsamples (see online supplemental figure 2). In the second model, regarding the HAT, those individuals classified in 'low' trajectories had a significant risk of mortality: in the ≤1935 subsample, there was a significant risk of death from 76 to 94 years, and from 97 to 98; in the 1936–1945 subsample from 63 onwards; and in the >1945 subsample from 60 onwards. Those following a 'medium' HAT had a significant risk of death in the ≤1935 subsample intermittently from 76 to 86 years and at 98 years; in the 1936–1945 subsample from 65 onwards; and in the >1945 subsample from 62 onwards (see online supplemental figure 2).

Figure 1 shows the estimated cumulative coefficients calculated from the third model (including all variables) for each birth cohort. In this model, although the risk effect of COPD increases across age, it was rather non-significant (only a small effect in the ≤1935 subsample around 76 and 77 years old). In the case of the HAT, 'low' trajectories were associated with a higher risk of mortality in the case of the ≤1935 subsample (from 88 to 90). There was a significant mortality risk in the 1936–1945 and the >1945 birth cohorts (from 71 onwards and 60 onwards, respectively). 'Medium' HAT had only a significant effect in the 1936–1945 subsample, from 74 onwards. The interaction between COPD and HAT was assessed in the third model. A significant effect was only found in the model with the ≤1935 subsample. The interaction showed a significant effect (higher risk of death) for participants with COPD and a 'low' HAT, with the highest risk of death at the age of 75 and from 81 to 87.

Completed and detailed results of the fitting of Aalen's additive regression models are presented in online supplemental figure 3.

## DISCUSSION

We analysed the association of COPD with the risk of mortality and the moderating role of HAT in the SHARE study, a population-based cohort of middle-aged and older adults from 12 European countries who were followed up for 9 years. To account for potential cohort effects, we analysed the results separately in three groups: those born after 1945 (aged 50+), born between 1936 and 1945 (ages from 58 to 70 years old), and born in 1935 or earlier (ages from 69 to 104 years old).

Our findings show that COPD increased the risk of mortality in the three birth cohorts. However, this association was no longer significant after adjusting for demographic and economic variables, presence of other NCDs

**Table 1** Main characteristics of the sample broken down by year of birth

| Characteristics | Years of birth cohort | | | P value* |
| --- | --- | --- | --- | --- |
| | ≤1935 (N=9738) | 1936–1945 (N=9254) | >1945 (N=9865) | |
| Female, n (%) | 5407 (55.50) | 4879 (52.70) | 5382 (54.60) | <0.001 |
| Age, mean (SD) | 74.40 (5.94) | 61.30 (2.95) | 52.10 (2.58) | <0.001 |
| Marital status, n (%) | | | | <0.001 |
| Single | 452 (4.64) | 468 (5.06) | 638 (6.47) | |
| Married | 5832 (59.90) | 7343 (79.30) | 8051 (81.60) | |
| Divorced | 338 (3.47) | 603 (6.52) | 950 (9.63) | |
| Widowed | 3646 (37.40) | 1203 (13.00) | 531 (5.38) | |
| Education level, n (%) | | | | <0.001 |
| Less than primary | 994 (10.20) | 417 (4.51) | 246 (2.49) | |
| Primary | 3895 (40.00) | 2513 (27.20) | 1604 (16.30) | |
| Secondary | 3734 (38.30) | 4570 (49.40) | 5520 (56.00) | |
| Tertiary | 1115 (11.40) | 1754 (19.00) | 2495 (25.30) | |
| Wealth quintiles, n (%) | | | | <0.001 |
| First (worst) | 1798 (18.50) | 801 (8.66) | 687 (6.96) | |
| Second | 2479 (25.50) | 1380 (14.90) | 960 (9.73) | |
| Third | 2185 (22.40) | 1895 (20.50) | 1481 (15.00) | |
| Fourth | 1721 (17.70) | 2247 (24.30) | 2494 (25.30) | |
| Fifth (best) | 1555 (16.00) | 2931 (31.70) | 4243 (43.00) | |
| Region, n (%) | | | | 0.049 |
| Northern Europe | 1557 (16.00) | 1483 (16.00) | 1473 (14.90) | |
| Western Europe | 4842 (49.70) | 4676 (50.50) | 5090 (51.60) | |
| Southern Europe | 3339 (34.30) | 3095 (33.40) | 3302 (33.50) | |
| Healthy ageing trajectories, n (%) | | | | <0.001 |
| High | 3073 (31.60) | 4620 (49.90) | 5962 (60.40) | |
| Medium | 4713 (48.40) | 3491 (37.70) | 3050 (30.90) | |
| Low | 1952 (20.00) | 1143 (12.40) | 853 (8.65) | |
| Physical activity, n (%) | 4934 (50.70) | 7107 (76.80) | 8283 (84.00) | <0.001 |
| Ever smoked, n (%) | 3910 (40.20) | 4572 (49.40) | 5543 (56.20) | <0.001 |
| Diseases, n (%) | | | | |
| Diabetes | 1821 (18.70) | 1628 (17.60) | 1132 (11.50) | <0.001 |
| Hypertension | 5334 (54.80) | 4647 (50.20) | 3500 (35.50) | <0.001 |
| Joint disorders | 3914 (40.20) | 3018 (32.60) | 2407 (24.40) | <0.001 |
| Asthma | 703 (7.22) | 550 (5.94) | 497 (5.04) | <0.001 |
| COPD | 1214 (12.50) | 886 (9.57) | 622 (6.31) | <0.001 |
| Myocardial infarction | 3022 (31.00) | 1751 (18.90) | 957 (9.70) | <0.001 |
| Stroke | 1119 (11.50) | 565 (6.11) | 332 (3.37) | <0.001 |
| Depression | 4340 (44.60) | 3387 (36.60) | 3444 (34.90) | <0.001 |
| Death, n (%) | 1642 (16.90) | 451 (4.87) | 204 (2.07) | <0.001 |

Household income was divided into five quintiles (the first indicating the lowest income). Marital status 'married' category included 'currently married or cohabiting' and 'divorced' included 'divorced or separated'.
*Based on t-tests for numerical variables and $\chi^2$ tests for categorical variables.
COPD, chronic obstructive pulmonary disease.;

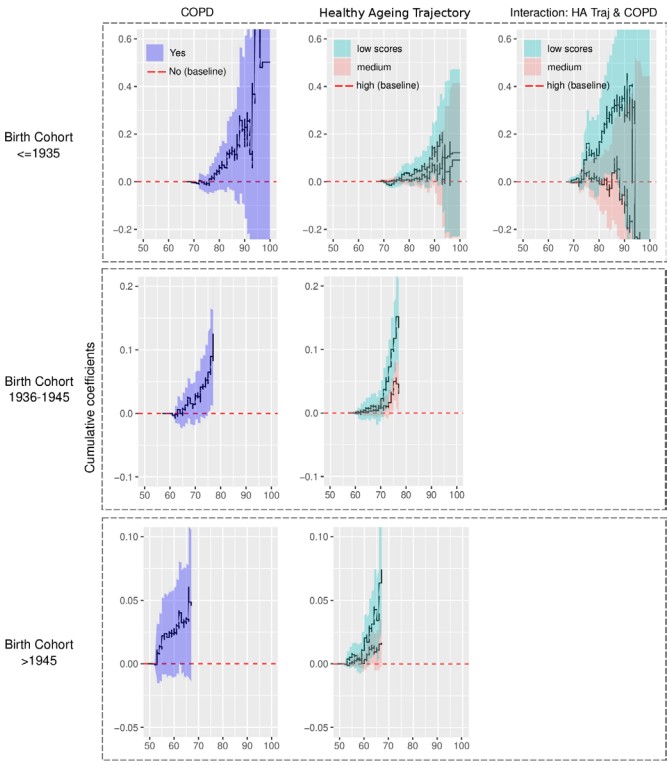

Models adjusted by sex, education, wealth, marital status, region, depression, diabetes, hypertension, joint disorders, asthma, myocardial infarction, and stroke:

**Figure 1** Cumulative excess risk of mortality associated with COPD, HAT by birth cohort and their interaction in the oldest birth cohort. All models were adjusted by sex, education, wealth, marital status, region, depression, diabetes, hypertension, joint disorders, asthma, myocardial infarction and stroke. COPD, chronic obstructive pulmonary disease; HAT, healthy ageing trajectories.

and depression, and HAT. In line with previous research, the study of mortality in patients with COPD is quite cumbersome, and multiple variables may play a role in this association. For example, lung cancer and COPD mortality were assessed, including several variables (residential characteristics, marital status, education, health insurance and family income) in a research study based on The National Longitudinal Mortality Study in the United States.[42] They found that COPD mortality rates were highest among 65–74 years old, in males and non-Hispanic whites.[42] The results concerning the periods are consistent with those we found before adjustment, suggesting the existence of a period of increased risk of mortality in patients with COPD. In another study based on The National Mortality Database of Statistics Canada, the mortality related to COPD varied by age, sex, birth cohort and the province.[43] In that study, the mortality risk attributed to COPD decreased in male and female cohorts born after 1920 to 1924, whereas between 1971 and 1983 the mortality ratios were stable.[43] Thus, performing the analyses considering different birth cohorts seems appropriate since exposure to risk factors for COPD such as tobacco consumption or occupational pollution might greatly vary across birth cohorts. Moreover, previous

studies on the risk of COPD mortality have reported differences in age, sex, birth cohort, location, household income, education and marital status.[42–45] Thus, the study of mortality associated with COPD needs to account for the potential confounding effects of these risk factors.

One potential confounder is the region of residence, as indicated in previous studies.[46] Despite not being the focus of our study, we identify that living in Western or Southern Europe had a protective effect on the risk of all-cause mortality, compared with Northern Europe (Denmark and Sweden). Similarly, Blanco *et al* found a lower mean COPD prevalence in Southern Europe (10.8%) compared with Northern Europe (11.5%). However, variations in COPD prevalence were also found among countries of the same European region.[12] In Northern Europe, it was higher in Denmark (ranging from 12% to 25%) than in Sweden (ranging from 2% to 20%); whereas in Southern Europe, Italy showed higher prevalence (ranging from 12% to 23%), than in Spain (from 7% to 10%).[12] The greater COPD prevalence and its associated mortality risk in Denmark could be a consequence of a very high smoking prevalence in the past five decades, resulting in the highest COPD prevalence in the western world.[47] This heterogeneity among countries and regions might suggest the need for a better understanding of the underlying mechanisms.

Regarding the HAT, our results seem to confirm that participants (from different birth cohorts) with 'low' and 'medium' HAT (ie, worse health status) have a higher risk of mortality compared with those classified into 'high' HAT. This effect remains after adjusting for covariates, although in the case of the 'medium' trajectories, only a significant effect was found in the 1936–1945 birth cohort (constituted by people aged 58–70 years old). According to our results, 'low' trajectories seem to discriminate a poorer health status in a better way and to predict mortality, even after adjusting for confounders. Previous studies examined the connection between healthy ageing and mortality, although using different indicators.[48 49] In a South Brazilian population-based cohort, researchers differentiated between normal ageing and successful ageing (defined as a good state of health, a complete absence of functional disability and mood changes, and no cognitive impairment).[48] They detected that successful agers had lower mortality rates, and the normal agers had a higher risk for mortality.[48] These results may be extrapolated to our 'low' and 'high' HAT, being the last the equivalent to 'successful ageing'. Another study used The Healthy Ageing Index (HAI) as a summary measure of physiologic ageing[49] composed of cardiovascular, lung, cognitive, metabolic and kidney function markers. In that study, HAI scores tended to increase with age (meaning worse healthy ageing) and predicted mortality from a given time point.[49] Hence, composite measures of ageing seem to be powerful tools to predict mortality and identify individuals at a higher risk.

One of the main results from our study is that the association between COPD and risk of mortality depended

on the HAT of the oldest participants (ie, born ≤1935). Individuals with COPD and a 'low' trajectory of healthy ageing were more likely to die at the age of 75 years old and from 81 to 87, compared with people with COPD and a 'medium' or 'high' HAT. The healthy ageing scale covers several domains (vitality, sensory skills, mobility, cognition and ADL/IADL) and could negatively affect those patients with worse COPD symptoms.[7–10] The fact that these results were found only in the oldest subsample may be related to the course of the disease since COPD is a progressive disease, and exacerbations and hospitalisations are particularly common among older individuals.[50] Our results point out temporary spaces where older patients with COPD with a 'low' HAT are at higher risk of mortality. Thus, future efforts should be concentrated on those aged 75 years old and from 81 to 87.

Few studies have analysed the relationship between health status in patients with COPD to the best of our knowledge. These studies were based on self-reported perceived health status assessed through the SF-12 questionnaire, a generic instrument to evaluate physical and mental health.[51 52] The main finding in one of these studies that used data from the BOLD project was that COPD severity was an important determinant of health status (more severity linked to poorer health status).[51] Although these studies considered the health status of people with COPD, we have not found any study that used a composite measure of healthy ageing as we have done. An integrated measure assessing intrinsic capacity and functional ability could be a useful tool in daily clinical practice for patient prognosis, as well as a mortality predictor, and for the creation of future public health strategies addressing COPD patients' needs.[21] While it is true that other composite tools to predict COPD mortality are available (such as St George's Respiratory Questionnaire,[53] or the BODE index[54]), the healthy ageing scale is a comprehensive tool that could be applied not only to patients with COPD but also to patients with multimorbidity.

### Strengths and limitations

These findings should be interpreted in light of the following limitations. First, the presence or absence of COPD and NCDs was based on self-reported diagnostics. Thus they might be affected by measurement errors. Nevertheless, some authors sustain self-reported diagnostics as a well-established method for measuring NCDs in population-based studies.[55] Second, we made some assumptions in terms of age of diagnosis. Due to the high percentage of missingness (48%) in the age of the first NCD diagnosis, we selected the age of the earliest diagnosis of each NCD within the five waves. That is, we coded the age of the participant in the wave he/she reported the first time having some of the included diseases. Despite being an assumption, there are only 2 years between each wave in the SHARE study. Thus, we believe that there is not a significant impact on our conclusions. Thirdly, we split the sample into three birth cohorts when performing

the analyses, and we reported the mortality risk in each group. By doing so, we captured potential cohort effects which people from different birth cohorts can be influenced by different exposure to COPD-related risk factors that contribute differently to mortality, as the different trends in smoking prevalence. For each birth cohort, the survival analysis can be focused according to the years of the interview or according to the age of the participants. We finally decided to do it according to the age of the participants because working with time-varying variables and without the assumption of proportional risks, the fluctuations in mortality risk according to age could be better interpreted. However, this introduces a problem of left truncation since the age range observed for each participant is different, although we took this into account in the additive regression model. Fourth, another issue is that the age range observed for each birth cohort is also different so that the excess cumulative risk curve starts at the first observed age. Therefore, the bias of the healthy participant in the first wave of the study means that there is no significant excess risk in the first ages of observation. Fifth, we considered several variables that could affect mortality in COPD patients, such as the presence of other NCDs, ever smoked, the practice of vigorous physical activity and the role of HAT on mortality risk. However, other known factors with cumulative effects on COPD, such as long-term smoking and physical activity or lung function data,[50] were not available in the study. Thus we could not control for their potential confounding effect. Alongside these limitations, this study had some strengths. First, the analyses were performed in different birth cohorts (>1945, 1936–1945 and ≤1935) to assess differences in mortality risks related to societal changes, such as lifestyle behaviours and occupation trends. Second, we used a novel measurement scale of healthy ageing, including intrinsic capacity and functional ability variables. Compared with the use of different health indicators separately, we believe that using an integrated and reliable measure of health status is a powerful tool to predict the mortality risk of the participants. Third, the calculation of Aalen additive hazards models rather than Cox models allowed the inclusion of time-variant variables in the analyses.

### CONCLUSION

COPD is a costly and preventable disease that has large-scale implications for patients' quality of life and society in general.[56 57] Our findings suggest that the association between COPD and the risk of mortality in the general population of middle-aged and older adults might be explained by the presence of other risk factors. However, for older people with COPD (ie, aged 69 or older), having a poor trajectory of healthy ageing might compromise their survival. Especial attention should be paid to these patients, with the healthy ageing scale as a suitable tool identifying older patients with COPD at high risk of mortality.[20]

**Author affiliations**
[1]Research, Innovation and Teaching Unit, Parc Sanitari Sant Joan de Déu, Sant Boi de Llobregat, Catalunya, Spain
[2]Department of Medicine, Universitat de Barcelona, Barcelona, Catalunya, Spain
[3]Centro de Investigación Biomédica en Red de Salud Mental (CIBERSAM), Instituto de Salud Carlos III, Madrid, Spain
[4]Serra Húnter fellow, Department of Statistics and Operations Research, Polytechnic University of Catalonia, Barcelona, Catalunya, Spain

**Acknowledgements** THE SHARE study is funded by the European Commission through FP5 (QLK6-CT-2001–00360), FP6 (SHARE-I3: RII-CT-2006–0 62 193, COMPARE: CIT5-CT-2005–0 28 857, SHARELIFE: CIT4-CT-2006–0 28 812) and FP7 (SHARE-PREP: No 211909, SHARE-LEAP: No 227822, SHARE M4: No 261982). Additional funding from the German Ministry of Education and Research, the Max Planck Society for the Advancement of Science, the US National Institute on Aging (U01_AG09740-13S2, P01_AG005842, P01_AG08291, P30_AG12815, R21_AG025169, Y1-AG-4553–01, IAG_BSR06-11, OGHA_04–064, HHSN271201300071C) and from various national funding sources is gratefully acknowledged (see www.share-project.org).

**Contributors** IB-M: Participated in database management, drafted the paper, carried out the statistical analyses, and worked on the interpretation of data. She also gave final approval of the version to be published and agreed to be accountable for all aspects of the work in ensuring that questions related to the accuracy or integrity of any part of the work are appropriately investigated and resolved; AS-N: Participated in the study design, database management, carried out the statistical analyses, gave statistical support and critical revision of the paper. He also gave final approval of the version to be published and agreed to be accountable for all aspects of the work in ensuring that questions related to the accuracy or integrity of any part of the work are appropriately investigated and resolved; DF: Participated in the statistical support and critical revision of the paper. He also gave final approval of the version to be published and agreed to be accountable for all aspects of the work in ensuring that questions related to the accuracy or integrity of any part of the work are appropriately investigated and resolved; JMH: Participated in the acquisition of data, and critical revision of the paper. He also gave final approval of the version to be published and agreed to be accountable for all aspects of the work in ensuring that questions related to the accuracy or integrity of any part of the work are appropriately investigated and resolved; BO: Participated in the critical revision of the paper. She also gave final approval of the version to be published and agreed to be accountable for all aspects of the work in ensuring that questions related to the accuracy or integrity of any part of the work are appropriately investigated and resolved.

**Funding** This work was supported by the 5-year Ageing Trajectories of Health: Longitudinal Opportunities and Synergies (ATHLOS) project. The ATHLOS project has received funding from the European Union's Horizon 2020 research and innovation programme under grant agreement No 635 316. BO's work is supported by the Miguel Servet Programme (CP20/00040), funded by Instituto de Salud Carlos III and cofunded by European Union (ERDF/ESF, 'Investing in your future'). DF's work has been supported by Marsden grant E2987-3648 administrated by the Royal Society of New Zealand) and by grant 2017 SGR 622 (GRBIO) administrated by the Departament d'Economia i Coneixement de la Generalitat de Catalunya (Spain).

**Competing interests** None declared.

**Patient consent for publication** Not required.

**Ethics approval** Ethical approvals for waves from 1 to 3 were granted by the Ethics Committee of the University of Mannheim. For waves 4 and 5, the SHARE projects were reviewed and approved by the Ethics Council of the Max-Planck Society. All data were anonymised and EHR confidentially was respected in accordance with national and international law.

**Provenance and peer review** Not commissioned; externally peer reviewed.

**Data availability statement** Data are available in a public, open access repository. The original data of the Survey of Health, Ageing and Retirement in Europe-SHARE is available on the official website (http://www.share-project.org/home0.html). R codes for harmonising the healthy ageing scale is available on https://athlos.pssjd.org/study/share-hs.

**ORCID iDs**
Ivet Bayes-Marin http://orcid.org/0000-0002-3816-5244
Albert Sanchez-Niubo http://orcid.org/0000-0003-0309-181X
Daniel Fernández http://orcid.org/0000-0003-0012-2094
Josep Maria Haro http://orcid.org/0000-0002-3984-277X
Beatriz Olaya http://orcid.org/0000-0003-2046-3929

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
