## [Reviewer comments · BMJ Open]

ARTICLE DETAILS

TITLE (PROVISIONAL)	Risk of all-cause mortality associated with chronic obstructive pulmonary disease and the role of healthy ageing trajectories: A population-based study of middle-aged and older adults.
AUTHORS	Bayes-Marin, Ivet; Sanchez-Niubo, Albert; Fernández, Daniel; Haro, Josep; Olaya, Beatriz

VERSION 1 – REVIEW

REVIEWER	Sciriha, Anabel University of Malta
REVIEW RETURNED	24-Mar-2021

GENERAL COMMENTS	A very interesting investigative study which looks at the incidence and mortality levels of such patients in Europe. In the introduction to your study, maybe some more emphasis about this should be made and maybe offer reasons for choosing Europe. In the Introduction sentence in lines 23-27 would require a reference. Same goes for sentence in line 31-33. In both the results and discussion, one statistically significant changes are discussed, included the statistical data would be recommended. Well done
---

REVIEWER	Axson, Eleanor Imperial College London, National Heart and Lung Institute
REVIEW RETURNED	23-Apr-2021

GENERAL COMMENTS	Minor Comments: 1. The authors may consider discussing how the lack of lung function data, long-term smoking data, and long-term physical activity data could impact their results -- although it is proxied through the functional abilities in the ageing scale, this would be an interesting discussion for the COPD community.
---

VERSION 1 – AUTHOR RESPONSE

Reviewer #1 (Dr. Anabel Sciriha, University of Malta)

COMMENT: "In the introduction to your study, maybe some more emphasis about this should be made and maybe offer reasons for choosing Europe"

RESPONSE: First of all, we thank the reviewer for the positive general appraisal of our study. We decided to analyse SHARE data, which includes European countries, because, according to the literature, there is a paucity of data about COPD prevalence and associated death rates in Europe and is expected to become the third leading cause of death by 2030. Moreover, the Burden of Lung Disease (BOLD) project and the Latin American Project for the Investigation of Obstructive Lung Disease (PLATINO) were performed in China and Turkey, and five Latin American countries, respectively; but homogeneous data (i.e., similar methodology and diagnostic criteria) is required to estimate COPD prevalence and associated mortality in Europe. In that manner, we considered that the SHARE study fits well for accomplishing our research objectives as it is a cross-national European database with 5 waves of follow-up. To clarify these points for the readers, we added two sentences in the introduction (in bold) and a reference, which we also included in the reference list.

“COPD is expected to become the third leading cause of death by 2030(3). The growing burden of COPD is a consequence of the population ageing and the continued use of tobacco, which is considered its main risk factor(4).”

“Despite being a growing public health concern, there is a lack of epidemiological data about the prevalence and distribution of COPD(6,12). The paucity of information on COPD prevalence and incidence is partly due to differences in the methods used for its diagnosis and classification, often being underestimated(6). These differences in the assessment methods and definitions have also hampered the comparison of COPD prevalence and impact across countries. Both The Burden of Lung Disease (BOLD) project(4) and the Latin American Project for the Investigation of Obstructive Lung Disease (PLATINO)(13) were developed to map the COPD prevalence using the same methodology in different countries. Those studies were performed in China and Turkey, and five Latin American countries, respectively. Nevertheless, there is a lack of integrated and updated estimates of COPD prevalence, similar to BOLD and PLATINO initiatives, and information regarding patient’s quality of life impact and associated mortality in Europe(2,14). Additionally, available studies suggest large differences across European countries in terms of prevalence rates of COPD and the associated death rates(15,16). In a systematic review, COPD prevalence ranged from 3% in Finnish women to 57% in Italian men and women(15). Some differences have also been found in COPD-related mortality across European countries and between men and women(16). Overall, regarding European countries, COPD-related mortality rates appeared to decline in men in most countries from 1995 to 2017, whereas mortality rates due to COPD increased in women from +2% per year in Austria to +4.2% or +4.8% per year in the Czech Republic and Hungary, respectively(16).

In that sense, a longitudinal integrated dataset that considers different European countries could be particularly useful in the study of the risk of mortality associated with COPD in different European countries. In particular, using a cross-national panel database may prevent possible heterogeneities arising from differences in survey methodologies, diagnostic criteria, and population structure(17). The study of the mortality risk in COPD patients, as well as its association with several variables related to health and functioning, would allow identifying those vulnerable sectors of the population and the creation of preventive measures and interventions in diverse healthcare systems.”

References

17. Davies Adeloye, Stephen Chua, Chinwei Lee, Catriona Basquill, Angeliki Papan, Evropi Theodoratou, Harish Nair, Danijela Gasevic, Devi Sridhar, Harry Campbell, Kit Yee Chan, Aziz Sheikh, Igor Rudan and GHERG (GHERG). Global and regional estimates of COPD prevalence: Systematic review and meta-analysis. *J Glob Health*. 2015;5(2):020415.

COMMENT: “In the Introduction sentence in lines 23-27 would require a reference. Same goes for sentence in line 31-33.”

RESPONSE: We appreciate the reviewer's comment. Thanks for pointing it out. We included the corresponding references in these sentences.

"Additionally, available studies suggest large differences across European countries in terms of prevalence rates of COPD and the associated death rates(15,16). In a systematic review, COPD prevalence ranged from 3% in Finnish women to 57% in Italian men and women(15). Some differences have also been found in COPD-related mortality across European countries and between men and women(16). Overall, regarding European countries, COPD-related mortality rates appeared to decline in men in most countries from 1995 to 2017, whereas mortality rates due to COPD increased in women from +2% per year in Austria to +4.2% or +4.8% per year in the Czech Republic and Hungary, respectively(16)."

COMMENT: "In both the results and discussion, one statistically significant changes are discussed, included the statistical data would be recommended".

RESPONSE: This is a relevant comment about methodology. We understand that the reviewer is asking for including probability measures (i.e., a p-value and/or confidence intervals) as a part of the results and the discussion with the aim of determining statistically significant changes when we talked about all-cause mortality in individuals with COPD (before (significant) and after adjustment (non-significant)). In this case, we would like to clarify that we considered all the variables as time-varying variables, which implies that the estimated effect is neither constant nor proportional over time, which produces quite a large number of estimated effects. For that reason, the results when performing Aalen additive hazards models are typically depicted graphically, which is the option we took for our framework. However, there is the option of including a table in the supplementary material, showing the confidence intervals for each of the time units (age) in the three birth cohorts, but we are aware that this could be tedious for the readers. In any case, if the reviewer considers this inclusion relevant, we could include this table in the supplementary material.

Reviewer #2 (Dr. Eleanor Axson, Imperial College London)

COMMENT: "The authors may consider discussing how the lack of lung function data, long-term smoking data, and long-term physical activity data could impact their results -- although it is proxied through the functional abilities in the ageing scale, this would be an interesting discussion for the COPD community."

RESPONSE: That is a very interesting remark. Unfortunately, information regarding lung function data, long-term smoking data, and long-term physical activity were not available. Since we totally agree with the reviewer, we included this issue as a limitation of our study:

"Fifthly, we considered several variables which could affect mortality in COPD patients, such as the presence of other NCDs, ever smoked, the practice of vigorous physical activity and the role of healthy ageing trajectories on mortality risk. However, other known factors with cumulative effects on COPD, such as long-term smoking and physical activity or lung function data (52), were not available in the study and thus we could not control for their potential confounding effect".